# Computational modeling of ketamine-induced changes in gamma-band oscillations: The contribution of parvalbumin and somatostatin interneurons

Jessie Rademacher[1], Tineke Grent-`t-Jong[1], Davide Rivolta[2], Andreas Sauer[3,4,5], Bertram Scheller[6], Guillermo Gonzalez-Burgos[7], Christoph Metzner[1,8,9], Peter J. Uhlhaas[1]*

**1** Department of Child and Adolescent Psychiatry, Charité Universitätsmedizin, Berlin, Germany, **2** Department of Education, Psychology, and Communication, University of Bari Aldo Moro, Bari, Italy, **3** Max Planck Institute for Brain Research, Frankfurt am Main, Germany, **4** Ernst Strüngmann Institute (ESI) for Neuroscience in Cooperation with Max Planck Society, Frankfurt am Main, Germany, **5** SRH University, Department of Applied Psychology, Heidelberg, Germany, **6** Department of Anesthesiology and Intensive Care Medicine, St. Josefs Hospital, Wiesbaden, Germany, **7** Department of Psychiatry, University of Pittsburgh, Pittsburgh, Pennsylvania, United States of America, **8** Neural Information Processing Group, Institute of Software Engineering and Theoretical Computer Science, Technische Universität Berlin, Berlin, Germany, **9** School of Physics, Engineering and Computer Science, University of Hertfordshire, Hatfield, United Kingdom

◉ These authors contributed equally to this work
* peter.uhlhaas@charite.de

## Abstract

Ketamine, an NMDA receptor (NMDA-R) antagonist, produces psychotomimetic effects when administered in sub-anesthetic dosages. While previous research suggests that Ketamine alters the excitation/inhibition (E/I)-balance in cortical microcircuits, the precise neural mechanisms by which Ketamine produces these effects are not well understood. We analyzed resting-state MEG data from n = 12 participants who were administered Ketamine to assess changes in gamma-band (30–90 Hz) power and the slope of the aperiodic power spectrum compared to placebo. In addition, correlations of these effects with gene-expression of GABAergic interneurons and NMDA-Rs subunits were analyzed. Finally, we compared Ketamine-induced spectral changes to the effects of systematically changing NMDA-R levels on pyramidal cells, and parvalbumin-, somatostatin- and vasoactive intestinal peptide-expressing interneurons in a computational model of cortical layer-2/3 to identify crucial sites of Ketamine action. Ketamine resulted in a flatter aperiodic slope and increased gamma-band power across brain regions, with pronounced effects in prefrontal and central areas. These effects were correlated with the spatial distribution of parvalbumin and GluN2D gene expression. Computational modeling revealed that reduced NMDA-R activity in parvalbumin or somatostatin interneurons could reproduce increased gamma-band power by increasing pyramidal neuron firing rate, but

**Data availability statement:** All code written in support of this publication is publicly available in the following respository: https://github.com/jessierademacher/Ketamine_E-I. The MEG and clinical data is not made publicly available as not explicit permission was obtained to share the data with third parties. For further inquiries please contact the Ethics Committee of the Medical Faculty, Goethe University Frankfurt (ethikkommission@unimedizin-ffm.de).

**Funding:** P.J.U. was supported by project MR/L011689/1 from the Medical Research Council (MRC), the ERA-NET project 01EW2007A, and through the Einstein Stiftung Berlin (A-2020-613). The funders had no role in study design, data collection and analysis, decision to publish, or preparation of the manuscript.

did not account for changes in the aperiodic slope. The results suggest that parvalbumin and somatostatin interneurons may underlie increased gamma-band power following Ketamine administration in healthy volunteers, while changes in the aperiodic component could not be recreated. These findings have implications for current models of E/I-balance, as well as for understanding the mechanisms underlying the circuit effects of Ketamine.

## Author summary

Previous research has shown that sub-anesthetic doses of Ketamine lead to increased excitation which could be mediated via effects on specific inhibitory GABAergic interneurons. In this study, we examined how the NMDA-R antagonist Ketamine influences the balance between excitation and inhibition in magnetoencephalography (MEG)-data and explored the role of different neuron types in mediating this effect. We found that Ketamine increased gamma-band activity and altered the slope of the aperiodic power spectrum in MEG-data. These effects were also linked to the expression of genes involved in regulating excitation and inhibition. Using a computational model, we simulated neural circuit activity following disruptions to specific neuron types and found that reducing two types of GABAergic interneurons (parvalbumin and somatostatin interneurons) produced changes in gamma-band oscillations similar to those observed with Ketamine in humans. These findings highlight the role of specific inhibitory neurons in the effects of Ketamine on neural circuits, offering new insights into how Ketamine alters the balance of excitation and inhibition.

## Introduction

Ketamine is an N-methyl-D-aspartate receptor (NMDA-R) antagonist which elicits, following sub-anesthetic administration to healthy volunteers, transient psychotic symptoms, such as hallucinations, delusions, and negative symptoms which overlap with the clinical presentation in schizophrenia patients [1,2]. An increasing body of work suggests that Ketamine-induced positive and negative symptoms as well as cognitive deficits involve alterations in the balance between excitation and inhibition (E/I-balance) in neural circuits [3,4]. E/I-balance has been shown to be essential for effective information processing in large-scale networks [5], and disruptions have been implicated to account for circuit and cognitive deficits in schizophrenia [6].

During normal brain functioning, excitation is balanced by inhibition produced by γ-amino butyric acid (GABA)ergic interneurons that inhibit pyramidal cells, thereby allowing for fluctuations in their excitability [5,7]. Since gamma-band oscillations (30–90 Hz) reflect the precise rhythmic interaction between excitatory and inhibitory neurotransmission, alterations in gamma-band oscillations might provide a non-invasive measure of changes in E/I-balance [8]. Moreover, recent evidence suggests

that the slope of the aperiodic, non-oscillatory component of brain activity may provide an additional marker for E/I-balance [9], as it has been shown to reflect the integration of excitatory and inhibitory synaptic currents [10] and to be associated with working memory performance [9,11], consciousness [12,13], and psychiatric disorders, such as depression, attention deficit disorder, and autism [14].

Gamma-band alterations have been implicated in circuit deficits in schizophrenia, and are thought to underlie cognitive impairments and certain clinical symptoms [15]. Specifically, reductions in the power of task-related gamma-band oscillations have been demonstrated during sensory and cognitive processes [16,17]. Moreover, there is evidence for alterations in resting-state activity and studies have shown both evidence for a reduction as well as an increase of gamma-band power [18]. Finally, preliminary evidence suggests that the aperiodic component might be altered in schizophrenia [19,20].

Among the mechanisms involved in maintaining E/I-balance, rhythmic inhibition by fast-spiking parvalbumin-expressing (PV+) interneurons is of particular interest as PV+ input onto pyramidal cells is crucial for generating gamma-band oscillations [21,22]. In addition, somatostatin-expressing (SST+) interneurons have also been shown to be involved in synchronizing gamma-band oscillations [23,24]. Finally, vasoactive intestinal peptide-expressing (VIP+) interneurons, which mainly target other GABAergic interneurons, including the PV+ and SST+ subclasses [25], may promote the disinhibition of pyramidal cells thus modulating oscillatory activity. Interestingly, the abundance of these different interneuron types varies across brain regions as indicated by distinct gene expression gradients [26].

Ketamine has been shown to have a profound effect on E/I-balance parameters through its action on glutamatergic synapses by non-competitively blocking NMDA-Rs [27]. While anesthetic doses of Ketamine have been shown to reduce excitation levels [28], sub-anesthetic doses increase the firing activity of pyramidal cells [29]. One possible mechanism is a preferential action of Ketamine on glutamatergic synapses on GABAergic interneurons, resulting in reduced inhibitory input to pyramidal neurons [30–32].

Increased disinhibition of neural circuits following Ketamine administration is consistent with altered gamma-band power in human EEG/MEG-recordings. Specifically, several studies have observed increased gamma-band power in both cortical and subcortical areas [33–35] but the precise circuit mechanisms that give rise to these changes remain unclear. Several rodent studies have found that NMDA-R antagonists, such as Ketamine and MK801, increase activity-levels of neurons with narrow spikes, hence putative fast-spiking interneurons [3,36]. However, excitatory synapses onto PV+ interneurons display a small NMDA receptor-mediated response to glutamate, relative to the aminomethylphosphonic acid (AMPA) receptor-mediated response [37,38]. In contrast, the NMDA-mediated contribution at synapses onto non-PV+ interneuron subtypes is larger [39,40]. Thus, by blocking NMDA receptors, Ketamine might affect cortical network activity via complex effects that involve multiple interneuron subtypes. Consistent with this idea, reduced activity following Ketamine administration has also been observed in SST+ interneurons [41,42]. Cichon et al. [43] observed that Ketamine decreased activity of PV+, SST+, and VIP+ interneurons in mice, while blockage of PV+ or SST+ interneurons was necessary for inducing the increase in pyramidal cell firing seen after Ketamine administration.

In the current study, we aimed to investigate the circuit mechanisms underlying the dysregulation of gamma-band activity following S-Ketamine administration in healthy volunteers, combining non-invasive MEG and computational modeling. Specifically, we examined the differential contribution of NMDA-Rs in PV+, SST+, and VIP+ interneurons towards alterations in gamma-band activity and in the slope of the aperiodic neural activity following subanesthetic Ketamine administration. We hypothesized that Ketamine administration would result in an increase in gamma-band power and a lower aperiodic slope (i.e., flatter power spectrum due to a shift towards more excitation), which would be mediated by blockage of NMDA-Rs on PV+ and SST+ interneurons.

## Results

### Demographic and PANSS data

We analyzed MEG-data from 12 healthy participants who received sub-anesthetic doses of Ketamine in a single-blind, randomized, placebo-controlled design. Resting-state MEG-data (5 minutes, eyes closed) was recorded before and during

continuous infusion of Ketamine or Placebo respectively. Demographic information, including age and sex information, as well as Positive and Negative Syndrome Scale (PANSS) [44] can be found in Table 1. PANSS scores in all six subscales were significantly increased after Ketamine administration (all $p$-values < 0.03).

### Gamma-band power

We computed gamma-band power (30–90 Hz) at sensor level and for virtual channels of 90 AAL atlas regions. Ketamine led to increased 30–90 Hz activity in the Ketamine vs. Placebo condition (Fig 1A) in a broad cluster of sensors, with a maximum effect over central and frontal sensors (cluster-$t$ = 1955.5, $p$ < 0.001, Cohen's $d$ = 2.95). Increased gamma-band power was localized across 79 cortical and subcortical regions with a mean effect size of $d$ = 3.66 (mean percent change: 21.05%) (see Table 2 and Fig 2A and 2B). The effects were highly consistent across participants (Fig 2C). A similar power increase could be observed in the periodic gamma-band (i.e., after subtraction of aperiodic activity; S2 File).

### Theta, alpha, and beta power

We additionally analyzed Ketamine-induced power changes in the theta (4–8 Hz), alpha (8–12 Hz), and beta (13–30 Hz) frequency ranges in both the non-separated as well as the periodic power spectrum. While no significant changes in theta and alpha power could be observed, Ketamine-induced decreases in the beta range were significant across several frontal and central brain regions both in the periodic and the non-separated data (for further discussion and comparison with computational modeling results, see S2 File).

### Aperiodic slope

We separated the periodic and aperiodic components of the MEG-data and calculated the slope of the aperiodic power spectrum between 7 and 80 Hz. The slope of the aperiodic component was flatter in the Ketamine condition compared to Placebo at sensor-level (see Fig 1B) (cluster-level $t$-value = -1557.7, $p$ < 0.001, Cohen's $d$ = -2.36), and across a wide range of brain regions, except for occipital cortex (Cohen's $d$ = -1.85, mean percent change from Placebo to Ketamine: -20.27%) (see Table 2 and Fig 2D and 2E).

**Table 1. Demographic and phenomenological information.**

|  | Placebo | Ketamine | Statistics |
|---|---|---|---|
| Age in years (range) | 29.6 (26-39) |  |  |
| Sex | 10 male, 2 female |  |  |
| **PANSS scores (SEM)** |  |  |  |
| Negative | 8.0 (0.6) | 13.3 (1.1) | $t(11)$ = 4.7, $p$ < 0.001 |
| Excitement | 5.5 (0.4) | 6.8 (0.6) | $t(11)$ = 2.5, $p$ = 0.028 |
| Cognitive | 5.5 (0.3) | 10.1 (0.7) | $t(11)$ = 6.5, $p$ < 0.001 |
| Positive | 4.1 (0.1) | 6.9 (0.5) | $t(11)$ = 6.2, $p$ < 0.001 |
| Depression | 5.7 (0.2) | 10.2 (0.4) | $t(11)$ = 9.9, $p$ < 0.001 |
| Disorganization | 3.1 (0.1) | 5.5 (0.6) | $t(11)$ = 4.1, $p$ = 0.002 |
| Total | 35.8 (1.0) | 58.6 (2.4) | $t(11)$ = 11.1, $p$ < 0.001 |

Positive and Negative Syndrome Scale (PANSS) scores were assessed at the completion of each session. Standard error of the mean (SEM) in brackets. Dependent-sample $t$-tests were used. Includes only participants whose data were included in the MEG analyses.

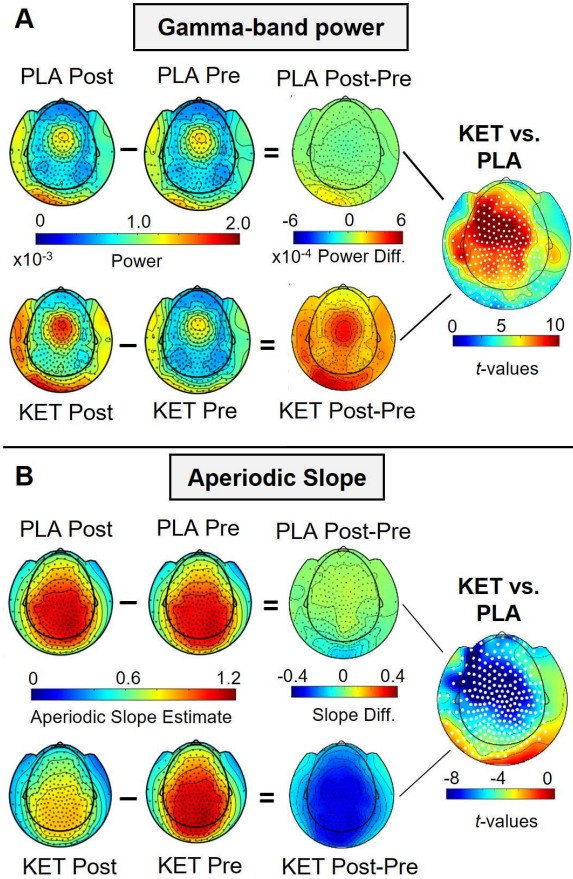

**Fig 1. Topography of gamma-band power and aperiodic slope effects.** (A) Depicted are grand-averaged (n = 12) topographies of averaged power estimates (30-90 Hz) for each condition (left panels), topographies of power difference values (post-condition, i.e., during continuous infusion, minus pre-condition, i.e., before administration; middle panels), and *t*-values of the statistical comparison (contrast of power difference values; right panels). Positive *t*-values indicate increased gamma-band power in the Ketamine compared to the placebo condition. White dots indicate sensors belonging to the cluster with significant effect. (B) Same as in (A) for the aperiodic slope effect. Negative *t*-values indicate a flatter slope in the Ketamine compared to the placebo condition. PLA = Placebo, KET = Ketamine.

To examine whether the slope change was driven by higher (> 30 Hz) vs. lower frequencies (< 30 Hz), analyses were repeated with slopes fitted between 7 and 30 Hz and between 30 and 80 Hz. While the slope of the lower-frequencies data did not differ between Ketamine compared to Placebo in any brain region, the slope for higher-frequencies was affected by Ketamine in several pre-frontal and central brain regions (see S2 Fig).

### Correlation with gene expression profiles

We calculated correlations between the topography of Ketamine-induced effects and the regional gene expressions of genes encoding the PV, SST, and VIP proteins (i.e., genes *PVALB*, *SST*, *VIP*) as well as genes encoding NMDA-R GluN2A-D subunits (i.e., *GRIN2A, GRIN2B, GRIN2C, GRIN2D)*.

*PVALB* gene expression levels across cortical areas were positively correlated with spatial differences in Ketamine-induced heightened gamma-band power (partial-*r* = 0.679, *p* = 0.006). The spatial differences in expression profiles of genes encoding PV and GluN2D correlated significantly with the spatial differences in Ketamine-induced negative slope change (i.e., correlated positively with a flatter slope; *PVALB*: partial-*r* = -0.671, *p* = 0.010, *GRIN2D*: partial-*r* = -0.426, *p* = 0.04; see Fig 3).

**Table 2. Overview of brain regions with significant Ketamine-induced increase in gamma-band power and aperiodic slope change.**

| | Significant brain regions | *t*-values | *p*-values |
|---|---|---|---|
| **Gamma difference** | | | |
| *Cortical* | LPreCG, RPreCG, LSFGdor, RSFGdor, LORBsup, RORBsup, LMFG, RMFG, LORBmid, RORBmid, LIFGoperc, RIFGoperc, LIFGtriang, RIFGtriang, LORBinf, RORBinf, LROL, RROL, LSMA, RSMA, LOLF, LSFGmed, RSFGmed, LORBsupmed, RORBsupmed, LREC, RREC, LINS, RINS, LACG, RACG, LDCG, RDCG, LPCG, RPCG, LPHG, RPHG, RLING, RSOG, LMOG, RIOG, LPoCG, RPoCG, LSPG, RSPG, LIPL, RIPL, LSMG, RSMG, LANG, RANG, LPCUN, RPCUN, LPCL, RPCL, LHES, RHES, LSTG, RSTG, LTPOsup, RTPOsup, LMTG, RMTG, LTPOmid, RTPOmid, LITG, RITG | 2.84 to 4.03 | 0.008 to 0.001 |
| *Subcortical* | LHIP, RHIP, LAMYG, RAMYG, LCAU, RCAU, LPUT, RPUT, LPAL, RPAL, LTHA, RTHA | 3.00 to 4.03 | 0.006 to 0.001 |
| **Slope difference** | | | |
| *Cortical* | LPreCG, RPreCG, LSFGdor, RSFGdor, LORBsup, RORBsup, LMFG, RMFG, LORBmid, LIFGoperc, LIFGtriang, LORBinf, LROL, RROL, LSMA, RSMA, LOLF, LSFGmed, RSFGmed, LORBsupmed, RORBsupmed, LREC, RREC, LACG, RACG, LDCG, RDCG, LPCG, RPCG, LPoCG, RPoCG, LSPG, RSPG, LIPL, RIPL, RMSG, LPCUN, LPCL, RPCL, LHES, RHES, LSTG, RSTG, RMTG, LTPOmid | 3.11 to 4.03 | 0.005 to 0.001 |
| *Subcortical* | RHIP, LAMY, LCAU, RCAU, LPUT, LPAL, LTHA, RTHA | 3.11 to 4.03 | 0.005 to 0.001 |

Abbreviations of brain regions according to AAL-atlas, for full labels see S1 File. Prefix: L = left, R = right.

## Computational modeling of NMDA-R dysfunction

Using a generic model of cortical layer 2/3 microcircuits developed by Yao et al. [45], we modeled the effects of NMDA-R dysfunction in either PV+, SST+, VIP+, or pyramidal neurons and examined whether the effects observed in human MEG-data after Ketamine administration could be simulated. The computational model consists of pyramidal neurons, PV+, SST+, and VIP+ interneurons, which were modeled to fit human data and connected in microcircuits (Fig 4B). The expected MEG signal from a volume of 1000 neurons (see Fig 4A) was simulated without manipulation (baseline model) and with the following reductions in NMDA-R conductance: PV+: 60%, 40%, 20%, 10%; SST+: 100%, 80%, 60%, 40%; VIP+: 80%; pyramidal: 80%, 40%; all neuron types simultaneously: 40% (see Fig 4B).

NMDA-R conductance reductions on PV+ or SST+ interneurons resulted in significant increases in gamma-band power (Fig 5A and 5B), with the magnitude of the power increase depending on the amount of reduction (see Table 3). Greater reductions on NMDA-Rs on SST+ interneurons resulted in higher increases in gamma-band power, with a complete removal of NMDA-Rs on SST+ interneurons leading to a mean power increase of 3.27 x 10e-21 (Cohen's $d = 4.16$). A comparable mean gamma-band power increase was observed with a 60% reduction of NMDA-Rs on PV+ interneurons (mean increase = 3.31 x 10e-21), yet this amount of NMDA-R conductance reduction in PV+ also caused considerably higher variance in power between simulations (see Table 3), diminishing the effect size (Cohen's $d = 2.41$). When comparing a 40% reduction in NMDA-R conductance between PV+ and SST+ interneurons, the mean power and effect sizes were of similar magnitudes (PV+-40%: mean power = 2.17 x 10e-21, Cohen's $d = 1.48$; SST+-40%: mean power = 2.10 x 10e-21, Cohen's $d = 1.36$). NMDA-R conductance reductions in pyramidal neurons, VIP+ interneurons, and in all neuron types simultaneously, did not result in an increase in gamma-band power (Fig 5C) (results of simulations with different combinations of neuron types affected are reported in S3 File and S3 Fig).

The slope of the aperiodic component of the simulated data, however, was not flatter after NMDA-R conductance reductions in any neuron type (all *p*-values > 0.44). While a power increase across the whole examined spectrum (3–90

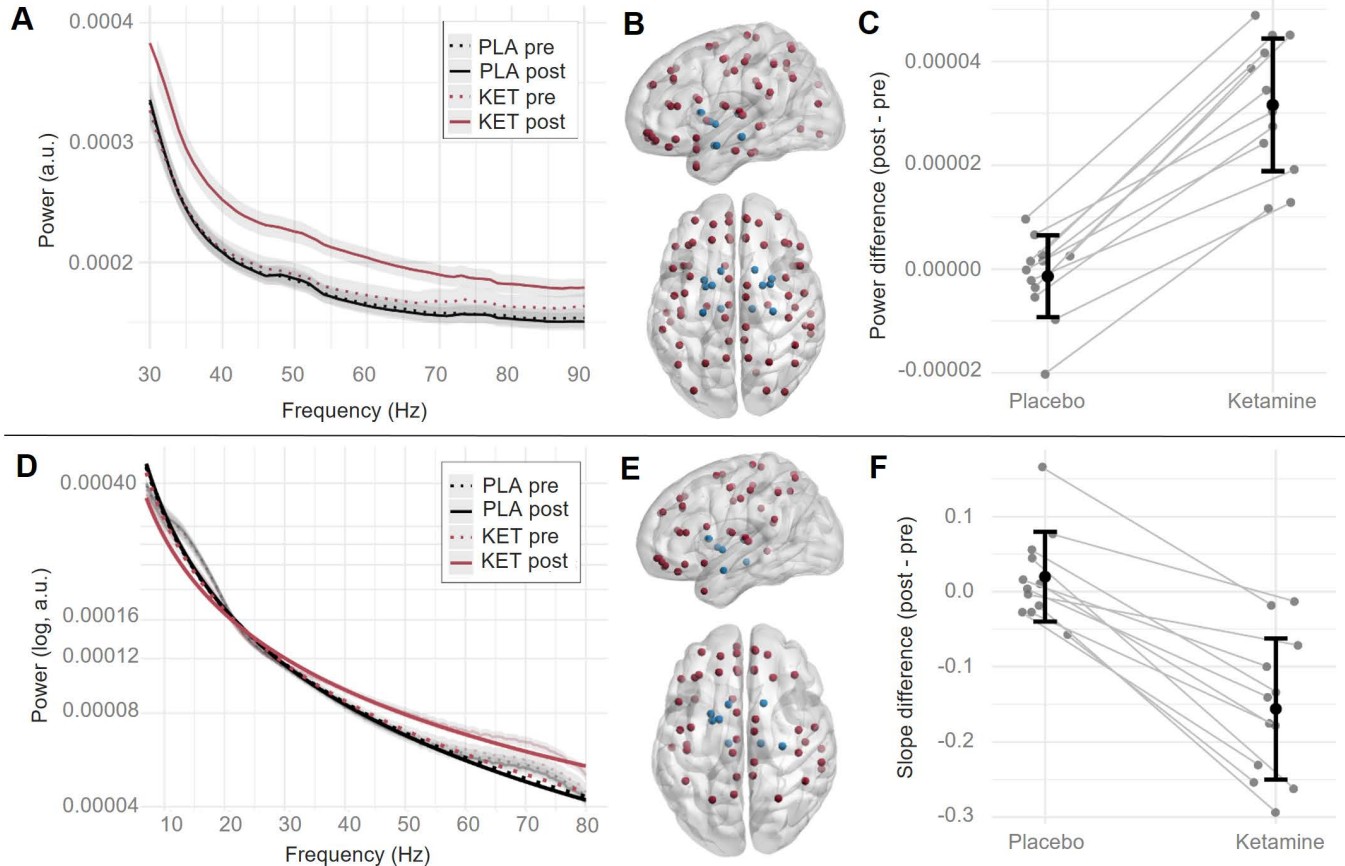

**Fig 2. Gamma-band power and aperiodic slope change at virtual channel level.** (A) Grand-averaged (n = 12) power spectrum in gamma range (30-90 Hz) per condition (black: Placebo, red: Ketamine, dotted line: pre infusion onset, straight line: post infusion onset), averaged across regions with significant gamma-band power change and across participants. Shaded envelope indicates standard error of the mean. (B) Centroids of cortical (red) and subcortical (blue) brain regions with significant gamma-band power change. Perspective from the left (upper figure) and above (lower figure) on a semi-transparent brain. Labels of these regions can be found in Table 2. (C) Gamma-band power difference values (post minus pre administration) for each participant (dark grey dots) averaged across regions with significant gamma power change, and averaged across participants (black dot) with standard deviation indication (black error bars). (D) Grand-averaged aperiodic fit (n = 12) per condition (black: Placebo, red: Ketamine, dotted line: pre infusion onset, straight line: post infusion onset), averaged across regions with significant slope change and across participants. The more transparent lines in the background show the aperiodic power spectrum in the respective conditions, averaged across regions with significant gamma-band power change and across participants, with the standard error of the mean as shaded envelope. (E, F) Same as (B, C) for the aperiodic slope.

Hz) of the aperiodic component of the data could be observed, the flattening of the aperiodic spectrum in MEG-data following Ketamine administration was not reproduced (see Fig 5D-F).

Modeling NMDA-R conductance reductions in pyramidal, PV+, SST+, and VIP+ neurons resulted in mean spike rate changes across neuron types (Fig 6). Elevated firing of pyramidal neurons (i.e., increased excitation) was observed in conditions with NMDA-R conductance reductions in PV+ interneurons or SST+ interneurons. Pyramidal cell spike rates were decreased when NMDA-Rs in VIP+ or pyramidal cells were affected.

## Discussion

The current study investigated the effects of Ketamine, a non-competitive NMDA-R antagonist, on E/I-balance in human MEG-data. Ketamine administration increased gamma-band power and flattened the aperiodic slope across brain regions

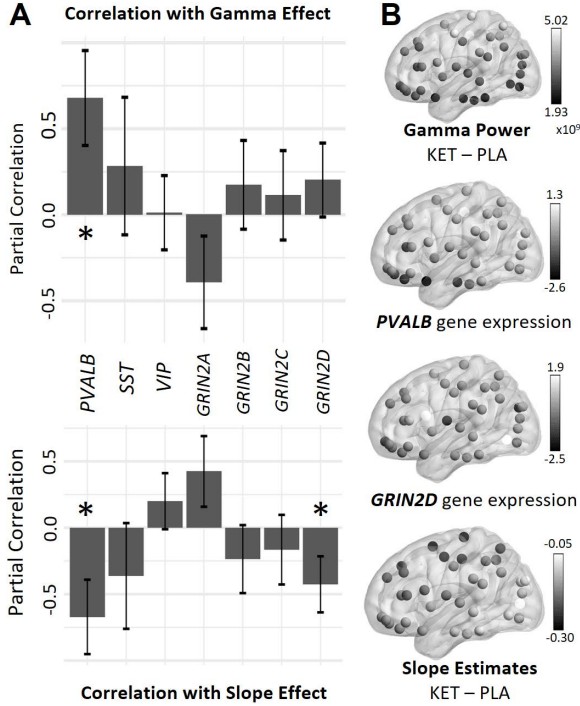

**Fig 3. Results of the partial correlations of different gene expressions with Ketamine-induced differences in gamma-band power and aperiodic slope.** (A) Partial correlations with standard error. * indicate significant partial correlations. Negative correlations with slope change indicate a correlation with a flatter slope after Ketamine administration. Positive correlations with gamma change indicate a correlation with elevated gamma-band power levels after Ketamine administration. (B) Spatial distributions of the gamma-band power and aperiodic slope difference estimates (Ketamine minus placebo) and the expressions of parvalbumin and GRIN2D genes. Dots depict centroids of the examined left-hemisphere regions according to the AAL atlas. Brighter dots indicate higher difference/gene expression values. Perspective from the left on a semi-transparent brain. KET = Ketamine, PLA = placebo, *PVALB* = parvalbumin gene, *SST* = somatostatin gene, *VIP* = vasoactive intestinal peptide gene, *GRIN2A* = gene for GluN2A NMDA receptor subunit, *GRIN2B* = gene for GluN2B NMDA receptor subunit, *GRIN2C* = gene for GluN2C NMDA receptor subunit, *GRIN2D* = gene for GluN2D NMDA receptor subunit.

and these changes correlated with expression levels across cortical areas for the PV-encoding (*PVALB*) and NMDA receptor subunit GluN2D-encoding (*GRIN2D*) genes. In our computational model, we found that the increase in gamma-band power could be reproduced by reducing NMDA-R levels in PV+ or SST+ interneurons. Interestingly, NMDA-R dysfunction on GABAergic interneurons, however, did not reproduce changes in the slope of Ketamine-induced changes in MEG-data.

The increase in gamma-band power after Ketamine administration is consistent with previous findings in rodents [46,47] and human resting-state EEG and MEG-data [33,48,49]. Increased gamma-band power has been linked to changes in E/I-balance, particularly to elevated levels of excitation [8,38,50]. Consistent with this hypothesis, a flatter aperiodic slope following Ketamine administration was observed in frontal, central as well as subcortical regions. Theoretical and empirical studies have shown that a flatter slope indexes elevated excitation [9,10]. Our results demonstrate that the change in slope was mainly driven by the aperiodic component in higher frequencies (> 30 Hz), which is consistent with previous observations of Gao et al. [10] that slope changes in these frequency ranges reflect E/I-changes.

Correlations between Ketamine-induced spectral changes and gene-expression data provided further evidence for the contribution of E/I-balance parameters. Changes in gamma-band power and the aperiodic slope induced by Ketamine correlated with the spatial distribution of *PVALB* gene expression, suggesting a preferential action via PV+ interneurons. The correlation of *SST* gene expression with changes in gamma-band power and aperiodic slope was also large, but did not reach statistical significance after correcting for spatial autocorrelation. Together with the simulation results showing that NMDA-R malfunction

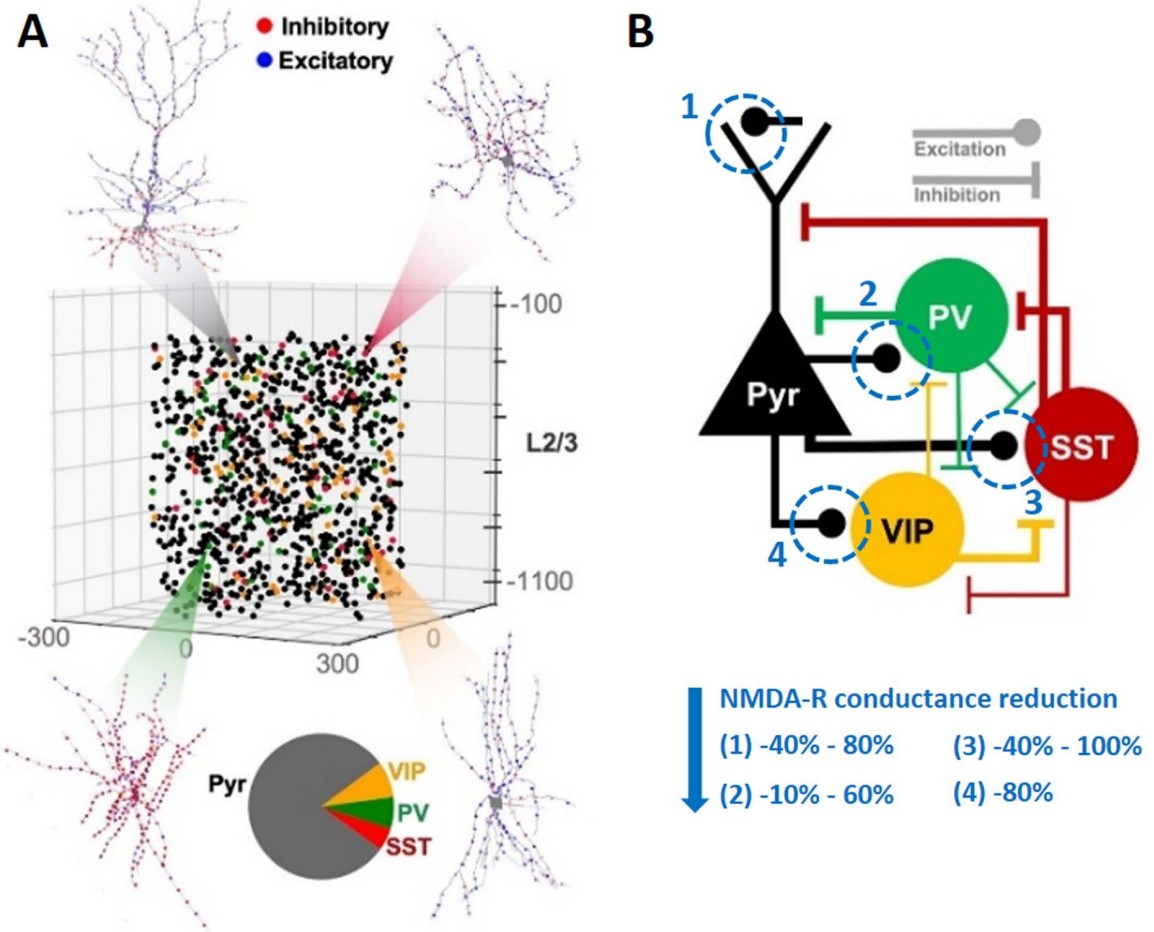

**Fig 4. Human cortical layer-2/3 model.** (A) Representation of the model with 1000 neurons, the neural morphologies of each modeled neuron type, and pie chart with the proportion of neuron types in the model (Pyramidal [Pyr] neurons 80%; somatostatin [SST] interneurons 5%; parvalbumin [PV] interneurons 7%; vasoactivate intestinal peptide [VIP] interneurons 8%). (B) Connectivity diagram of the microcircuit with main connections between neuron types. Blue circles indicate the site at which NMDA-R conductance was reduced (NMDA to [1] pyramidal cells, [2] PV interneurons, [3] SST interneurons, and [4] VIP interneurons). (A, B) Reprinted and adapted with permission from [45].

in SST+ interneurons reproduces increased gamma power as in human MEG data, these findings mayindicate that SST-related effects may exist but are more subtle than those associated with *PVALB* gene expression. Moreover, Ketamine-induced effects correlated with NMDA-R subunit gene expression. GRIN2D is expressed in about 60–80% of SST+ and PV+ interneurons in adult mice [51,52] with minimal expression in principal neurons [53,54]. Results on NMDA-R subunit-specific sensitivity to Ketamine have been inconsistent [55], but under physiological conditions, Ketamine may have more pronounced effects on GluN2C and GluN2D receptors compared to GluN2A and GluN2B receptors [56].

Using a computational model simulating human cortical layer-2/3 microcircuits, we were able to identify the contribution of specific GABAergic interneurons and their interactions with NMDA-Rs towards dysregulated gamma-band activity. Previous studies had shown that Ketamine inhibits fast-spiking [3,36] but also SST+ and VIP+ interneurons [41–43]. In all simulated conditions that elicited higher gamma-band power, pyramidal neuron firing rates were increased, consistent with prior observations [3]. Accordingly, dysregulated gamma-band activity following NMDA-R dysfunction may arise from an increased but desynchronized firing pattern of excitatory neurons due to reduced inhibitory input [50].

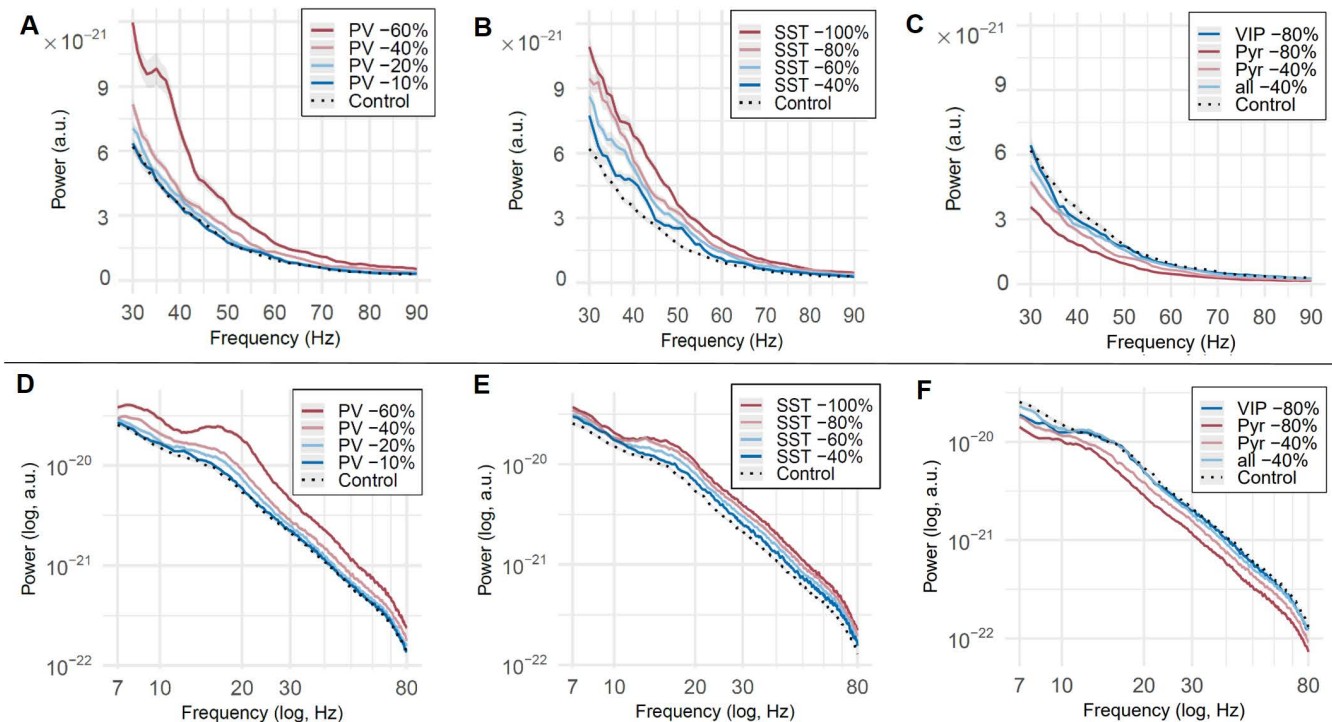

**Fig 5. Gamma-band power and aperiodic components of simulated data with NMDA-R reductions.** Panels (A-C) depict the averaged power spectrum of the fast-Fourier transformed data in the gamma-power range. Panels (D-F) show the aperiodic component of the power spectrum in log-log space. Shaded envelopes indicate standard error. Control condition (black, dotted line) without any manipulations. In the test conditions (straight, colored lines), NMDA receptors of (A, D) parvalbumin neurons, PV, (B, E) somatostatin neurons, SST, (C, F) vasoactive-intestinal peptide neurons, VIP, and pyramidal neurons, Pyr, and in all aforementioned neuron types simultaneously, were reduced by the indicated amount.

**Table 3. Gamma-band power increase in modeled NMDA-R dysfunction.**

| Condition | mean power (x10⁻²¹) | standard deviation (x10⁻²²) | *p*-value | *t*-value | Cohen's *d* | Percent change |
|---|---|---|---|---|---|---|
| baseline | 1.72 | 2.48 | | | | |
| PV -10% | 1.74 | 2.46 | 0.351 | -0.38 | 0.08 | 1.16% |
| PV -20% | 1.90 | 3.02 | < 0.001 | -3.24 | 0.65 | 10.47% |
| PV -40% | 2.17 | 3.59 | < 0.001 | -7.36 | 1.48 | 26.16% |
| PV -60% | 3.31 | 8.99 | < 0.001 | -12.03 | 2.41 | 92.44% |
| SST -40% | 2.10 | 3.09 | < 0.001 | -5.64 | 1.36 | 22.09% |
| SST -60% | 2.47 | 3.26 | < 0.001 | -13.02 | 2.61 | 43.60% |
| SST -80% | 2.82 | 4.31 | < 0.001 | -15.6 | 3.12 | 63.95% |
| SST -100% | 3.27 | 4.66 | < 0.001 | -20.80 | 4.16 | 90.12% |

Independent one-sided *t*-test between conditions with dysfunctional NMDA receptors in either parvalbumin (PV) or somatostatin (SST) interneurons in the indicated proportion of neurons and a baseline model without NMDA receptor manipulations. *P*-values are false-discovery-rate corrected for multiple comparison. Percent change from mean gamma-band power at baseline. N = 50 simulations per condition.

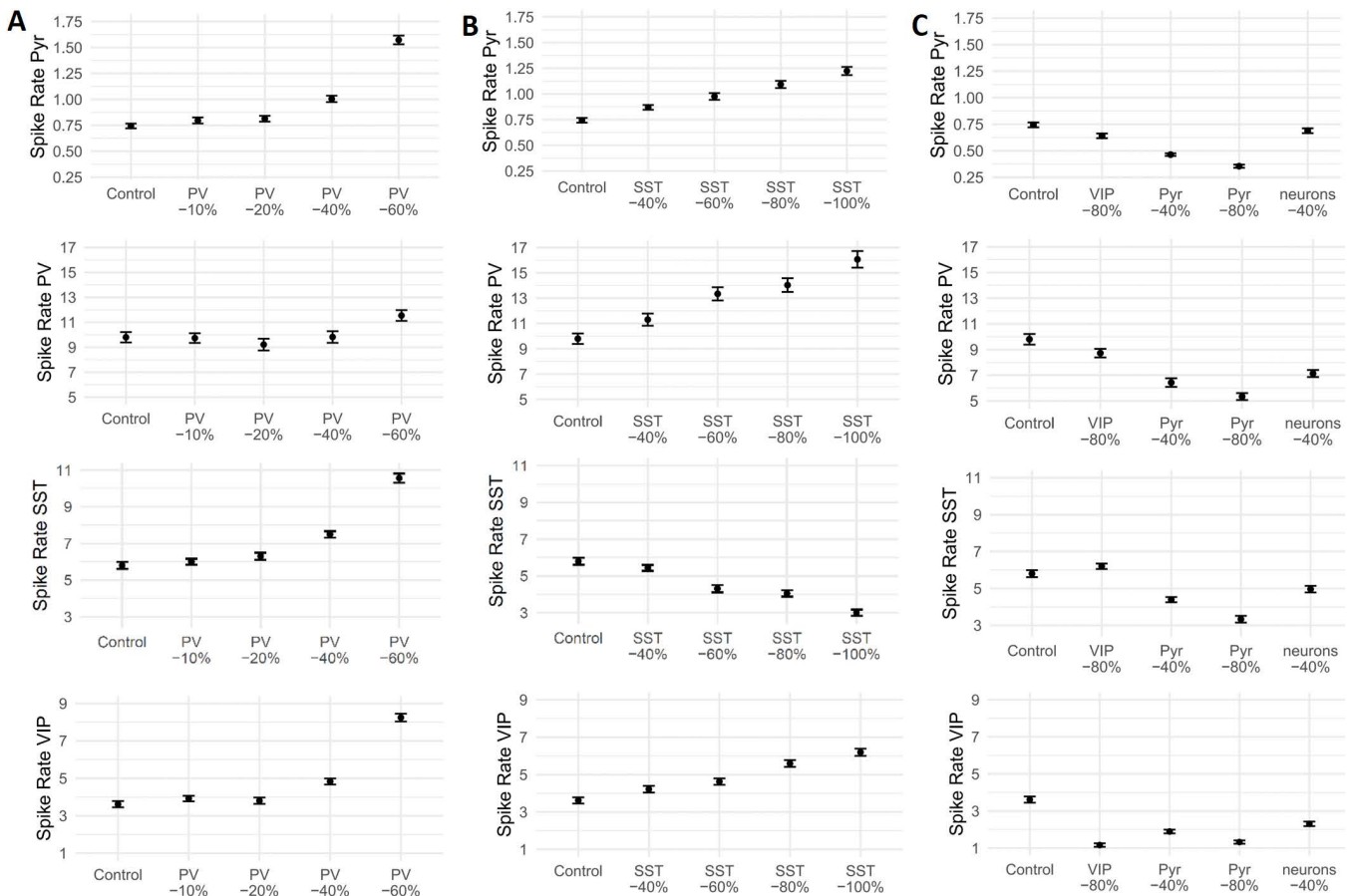

**Fig 6. Mean spike rates of different modeled neuron types with NMDA receptor reductions.** In different modeling conditions, NMDA receptors of (A) parvalbumin interneurons, PV, (B) somatostatin neurons, SST, (C) vasoactive-intestinal peptide neurons, VIP, or pyramidal neurons, Pyr, and in all aforementioned neuron types simultaneously, were reduced by the indicated amount. Control condition without any manipulations. Mean spike rate is given in Hz. Error bars indicate standard error of the mean.

Our data showed that NMDA-Rs on PV+ or SST+ interneurons were the main contributors towards increased gamma-band power following Ketamine administration, while reductions of NMDA-Rs on VIP+ and pyramidal cells did not increase gamma-band power. Reductions of NMDA-R activity in all neuron types did also not show a gamma-band increase, indicating that Ketamine might preferentially inhibit NMDA-Rs on specific neuron types. The specific effects of NMDA-Rs on PV+ and SST+ interneurons are consistent with the faster spike rate of PV+ and SST+ relative to VIP+ and pyramidal neurons [57,58] which is an important determinant for the generation of high-frequency oscillations [21]. A fast spike rate might lead to increased relief of NMDA-Rs from magnesium block, and could thereby enable Ketamine to inhibit these neuron types more easily [58]. However, there were also differences between simulations with different NMDA-R expression levels on PV+ and SST+ interneurons. Pronounced reductions of NMDA-Rs on PV+ interneurons distorted firing properties while comparable simulations with SST+ interneurons did not show such effects, indicating a greater sensitivity of PV+ interneurons towards the effects of NMDA-R hypofunction and highlighting the importance of PV+ interneurons for the maintenance of E/I-balance.

Moderate reductions of NMDA-Rs in both PV+ and SST+ interneurons resulted in comparable gamma-band power increases, indicating that both interneuron types are contributing towards E/I-balance in human microcircuits and might

underlie Ketamine's effects on gamma-band oscillations. However, computational modeling was not able to reproduce the changes in the aperiodic slope following Ketamine administration. Importantly, the interpretation of the slope as a measure of microcircuit E/I-balance has recently been challenged [59], suggesting that the aperiodic slope may be influenced by central states, such as arousal [60–62]. Furthermore, Ketamine is not only an NMDA-R antagonist but also targets dopaminergic, serotonergic and cholinergic receptors [55]. Accordingly, it is also conceivable that downstream effects of local E/I-imbalance, caused by Ketamine acting on NMDA-Rs of interneurons interact with Ketamine's effects on dopaminergic and cholinergic transmission to elicit slope changes. Furthermore, more complex mechanisms by which Ketamine might disrupt E/I-balance have been proposed, e.g., by acting on extra-synaptic NMDA-Rs of pyramidal neurons [58,63].

The current approach combining MEG-activity with computational modeling to explore the mechanisms of E/I-balance alterations is also relevant for understanding the biological basis of schizophrenia, which is characterized by prominent alterations in gamma-band oscillations [64] as well as alterations in excitatory and inhibitory neurotransmission [65]. However, it is currently unclear whether impairments in GABAergic interneurons, for example, are a primary dysfunction or whether these reflect a compensatory deficit towards impaired excitatory inputs [66]. Accordingly, advances in computational modelling could contribute towards disentangling competing circuit hypotheses [67].

The current study has several limitations. Although gamma-band power and aperiodic slope were used as distinct markers of E/I-balance, these measures are strongly correlated. It cannot be ruled out that changes in gamma-band power are driving the change in aperiodic slope in our MEG data, and that they might reflect overlapping aspects of the same underlying mechanism. Additionally, the computational model was restricted to cortical layer 2/3. While superficial layers are crucial for E/I-balance and are the main origin of gamma-band activity [68,69], it is not clear whether slope changes are specifically linked to layer 2/3 activity. The sample size of 12 participants was relatively small but the effects of Ketamine on gamma-band activity were highly robust and consistent across participants. Another limitation concerns the generic nature of the computational model, which was not optimized to simulate a particular brain region. Lastly, the baseline model assumed equal synaptic conductances from NMDA and AMPA receptors and in different (inter-)neuron types, which could lead to an overestimation of the impact of NMDA downregulation on specific cell types.

## Summary and implications

Our study provides novel evidence using computational modeling on the role of NMDA-Rs on different GABAergic interneurons towards dysregulated gamma-band oscillations following the administration of Ketamine. Firstly, our computational simulations recreated previous findings showing that Ketamine is associated with an increased firing in excitatory pyramidal cells. Secondly, we can show that both reduced NMDA-Rs on PV+ and SST+ interneurons may underlie the dysregulation of gamma-band activity in human MEG-activity following sub-anesthetic administration of Ketamine. However, while a Ketamine-induced change in the aperiodic slope was found in human MEG data, our computational modeling results suggest that NMDA-R dysfunction in cortical layer 2/3 interneurons do not account for this effect.

## Materials and methods

### Ethics statement

The study was approved by the ethical committee of Goethe University Frankfurt and carried out according to the Declaration of Helsinki. Participants gave written informed consent after complete description of the study and were monetarily rewarded for their participation.

The analysis plan of this project was registered under https://osf.io/u4f2y. Any deviations from the registered protocol will be reported along with a justification. Moreover, MEG-data from this study were previously published [35,70].

## Participants and experimental procedure

The sample consisted of 13 participants (2 females; mean age = 29.6). Only 12 participants were included in the analysis because of missing data. The study followed a single-blind, randomized, placebo-controlled crossover design. An initial bolus of 10 mg S-Ketamine (drug condition) or 10 ml sodium chloride (NaCl) 0.9% (Placebo condition) was administered, followed by a continuous intra-venous infusion of 0.006 mg S-Ketamine or NaCl 0.9% per Kg body weight per minute. The order of drug and Placebo conditions was randomized between participants.

Resting-state MEG activity was recorded for 8 minutes (4 minutes eyes closed, 4 minutes eyes open) three times on each session: 1) before bolus injection, 2) 5 minutes after the onset of the continuous infusion, and 3) 45 minutes later after completion of a visual grating task. Here, only the first two resting-state data sets (i.e., before bolus and 5 minutes after onset) of the eyes-closed condition were used. Results from this data-set were previously reported by Rivolta et al. [35].

Participants were excluded if they met a past or present axis I or II diagnosis (screened with SCID-II) or had a family history of psychosis.

## Neuroimaging

MEG data were acquired using a 275-sensors whole-head system (Omega 2005, VSM MedTech Ltd, BC, Canada) at a sampling rate of 600 Hz in a synthetic third order axial gradiometer configuration. A high-resolution anatomical MRI scan of each participant was acquired using a 3D magnetization-prepared rapid-acquisition gradient echo sequence in a 3T Siemens Trio scanner (160 slices, voxel size: 1x1x1 mm, FOV: 256 mm, TR: 2300 ms, TE: 3.93 ms). Markers placed on the nasion and pre-auricular points (bilaterally) were used to guide subsequent co-registration of the MEG data to the structural MRI scan.

## MEG data preprocessing and analysis

The Matlab toolbox 'FieldTrip', version 20221223 [71] was used for all MEG analysis. For figure generation, R (version 4.2.2; R Core Team, 2013) [72] with ggplot2 [73], FieldTrip, and BrainNet Viewer [74] were utilized. The data was segmented into 2s intervals with an overlap of 0.5s, down-sampled to 300 Hz, and notch-filtered to remove line-noise. The segments were visually inspected for muscle artifacts and SQUID jumps, and independent component analysis (ICA) was used to remove further artifacts. Trial numbers did not differ significantly between conditions (Placebo Pre: 225, Placebo Post: 224, Ketamine Pre: 224, Ketamine Post: 218; $F(3,44) = 1.09$, $p = 0.365$).

Analyses were conducted on both sensor and virtual channel level. We used the linearly constrained minimum variance (LCMV) beamformer approach [75] to reconstruct the MEG data from MNI source locations corresponding to centroids of the AAL atlas regions (90 regions; excluding cerebellum) [76]. For the estimation of spectral power, a Slepian-window multi-tapered Fast Fourier Transform (FFT) with a smoothing of 6 Hz was applied and the power estimates averaged between 30 and 90 Hz.

To separate the oscillatory and aperiodic components of the power spectrum, we applied the irregular resampling auto-spectral analysis (IRASA) algorithm [77] in a range between 3 and 90 Hz. For the calculation of the aperiodic slope, we fitted the logarithmic function $L(F) = b - \log_{10} \cdot F^{x}$ to the aperiodic component of the data in the frequency range F = [7,80] after log-transformation of the power estimates L. The exponent χ corresponds to the negative linear slope of the function in log-log space. An example fit can be seen in S1 Fig.

We tested for significant differences in the aperiodic slope and broadband gamma power between Ketamine condition (i.e., Ketamine post infusion onset – Ketamine pre infusion onset) and Placebo condition (i.e., Placebo post infusion onset – Placebo pre infusion onset) using Monte-Carlo permutation dependent t-test statistics with a cluster-based correction (sensor level) and false discovery rate (FDR) correction (virtual channel level). Two-sided t-tests with an alpha level of 0.05 were performed.

## Correlations between MEG-data and gene expression

We examined the topography of Ketamine-induced gamma-band and aperiodic slope effects in relationship to the regional gradient of expression of different E/I-related genes. To this end, we used data of the Allen Human Brain Atlas [78], (preprocessed by Burt et al. [26]) to assess expression of the genes encoding the PV, SST, and VIP proteins (i.e., genes *PVALB*, *SST*, *VIP*) as well as genes encoding NMDA-R GluN2A-D subunits (i.e., *GRIN2A, GRIN2B, GRIN2C, GRIN2D)*.

As only left-hemisphere gene data was available, we compared Ketamine-induced changes and gene expressions in 30 left-hemisphere AAL regions. Given the high correlations among the selected genes, we calculated partial correlations to isolate the effects of specific gene expressions. To prevent inflated *p*-values due to spatial autocorrelation (SA), we produced 1000 SA-preserving surrogate brain maps using the python package 'BrainSMASH' [79], using the default parameters, and calculated the statistics of the partial correlations based on this generative null model.

## Computational modeling

We used a generic model of a layer 2/3 (L2/3) cortical microcircuit developed by Yao et al. [45], which is a data-driven model of cortical L2/3 neurons based on mainly human data. It consists of 1000 neurons distributed in a 500 x 500 x 950 µm³ volume, 250–1,200 µm below the pia (see Fig 4A). Morphological, electrophysiological and anatomical (density of neurons and synaptic connectivity) features within the microcircuit model were fit to human data where possible and to rodent data otherwise, and show close matches with the experimental characteristics of the modeled populations [78,80–85].

The model was simulated using NEURON [86] and LFPy [87]. Background excitatory input representing cortical and thalamic drive was modeled so that the recurrent activity within the model shows realistic spontaneous firing rates for the different modeled cell types. Several models with reductions in NMDA-R conductance in PV +, SST +, VIP +, or pyramidal neurons were evaluated to systematically test the contribution of different expression levels of NMDA-Rs on GABAergic interneuron subtypes (PV +: -60%, -40%, -20%, -10%; SST +: -100%, -80%, -60%, -40%, VIP +: -80%, pyramidal: -80%, -40%; see Fig 4B). NMDA-R conductance reductions of more than 60% in PV+ interneurons resulted in unrealistic firing patterns and power spectra with strong oscillations at around 12 Hz and a power increases 30 times higher than in the control condition and were therefore not further evaluated. Additionally, simulations with NMDA-R conductance reductions of 40% in all four neuron types (pyramidal, PV +, SST +, VIP+) simultaneously were tested. To ensure the robustness of the finding with regard to the stochastic nature of the connectivity rules, we generated 50 randomized microcircuits with a duration of 3 s each, of which the first second was removed prior to analysis to avoid potentially disrupted onset activity.

To compare the simulated data to MEG-recordings, we modeled the expected MEG activity of the simulated neural activity using an infinite homogeneous volume conductor model [88] through the Python module LFPy. A fixed dipole origin was placed at the midpoint of L2/3 (-725 µm). The average signal from four sensors placed at the corners of a square with a side length of 1 cm located on the surface perpendicular to the computational model was then computed. Gamma-band (30–90 Hz) power and the slope of the aperiodic activity (modeled between 7 and 80 Hz) of the simulated data was analyzed in the same way as the MEG data described earlier.

To test which change in parameters resulted in a significant difference in gamma-band power and aperiodic slope compared to control conditions (i.e., no NMDA-R manipulation), independent-sample one-sided *t*-tests (informed by the direction of the effect in the MEG data; alpha-level = 0.05) were used, FDR-corrected for multiple comparisons.

Moreover, to test the effects of NMDA-R manipulations on the L2/3 neurons' firing properties, we calculated the averaged spike rates of pyramidal, PV +, SST +, and VIP+ neurons for each simulated condition.

## Supporting information

**S1 Fig. Example fit of the aperiodic component of one participant in LSFGmed.** (A) Power log-transformed. (B) Power and Frequency log-transformed.
(PNG)

**S2 Fig. Comparison of aperiodic slope changes in different frequency ranges.** (A) Upper panel: Power spectrum (log-transformed) of the aperiodic component per condition, averaged across regions with significant slope change and across participants, in a frequency range between 7 and 80 Hz. Lower panel: Centroids of cortical (red) and subcortical (blue) brain regions with significant slope change (slope fitted between 7 and 80 Hz). Perspective from the left and above on a semi-transparent brain. (B-C) Same as (A) but in a frequency range of 7–30 Hz (B) and 30–80 Hz (C).
(PNG)

**S3 Fig. Mean spike rates (A), gamma-band power (B) and aperiodic component (C) of simulated data with NMDA receptor reductions.** (A) Error bars indicate standard error of the mean. Control condition without any manipulations. In the test conditions, NMDA receptors of different combinations of parvalbumin neurons (PV), somatostatin neurons (SST), pyramidal neurons (Pyr) and vasoactive-intestinal peptide neurons (VIP) were reduced by the indicated amount. In the 'spike rate' condition, NMDA receptors were reduced proportional to the spike rate of that neuron type (NMDA receptor reductions: PV -40%, SST -20%, VIP -15%, Pyr -5%). In the 'interneurons' condition, NMDA receptors of PV, SST, and VIP interneurons were reduced by 40%. In the 'neurons' condition, NMDA receptors of all modeled neurons types (PV, SST, VIP, and pyramidal neurons) were reduced by 40%. (B) Averaged power spectrum of the fast-Fourier transformed data in the gamma-power range. (C) Aperiodic component of the power spectrum in log-log space. Shaded envelopes indicate standard error. Control condition (black, dotted line) without any manipulations. In the test conditions (straight, colored lines), NMDA receptors of PV, SST, VIP, and Pyr neurons, were reduced by the indicated amount.
(PNG)

**S1 File. AAL region abbreviations.**
(DOCX)

**S2 File. Power Differences in Different Frequency Ranges.**
(DOCX)

**S3 File. Statistical information on computational scenarios after reduction of NMDA-Rs in combinations of neurons.**
(DOCX)

## Author contributions

**Conceptualization:** Jessie Rademacher, Tineke Grent-'t-Jong, Davide Rivolta, Andreas Sauer, Bertram Scheller, Christoph Metzner, Peter J. Uhlhaas.

**Formal analysis:** Jessie Rademacher, Tineke Grent-'t-Jong, Christoph Metzner.

**Funding acquisition:** Peter J. Uhlhaas.

**Investigation:** Davide Rivolta, Andreas Sauer, Bertram Scheller, Peter J. Uhlhaas.

**Methodology:** Jessie Rademacher, Davide Rivolta, Andreas Sauer, Bertram Scheller, Guillermo Gonzalez-Burgos, Christoph Metzner, Peter J. Uhlhaas.

**Supervision:** Christoph Metzner, Peter J. Uhlhaas.

**Visualization:** Jessie Rademacher, Tineke Grent-'t-Jong.

**Writing – original draft:** Jessie Rademacher, Tineke Grent-'t-Jong, Christoph Metzner, Peter J. Uhlhaas.

**Writing – review & editing:** Jessie Rademacher, Guillermo Gonzalez-Burgos, Christoph Metzner, Peter J. Uhlhaas.

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
