## [Decision Letter · Decision Letter 0]

PCOMPBIOL-D-24-01769

Computational Modeling of Ketamine-Induced Changes in Gamma-Band Oscillations: The Contribution of Parvalbumin and Somatostatin Interneurons

PLOS Computational Biology

Dear Dr. uhlhaas,

Thank you for submitting your manuscript to PLOS Computational Biology. After careful consideration, we feel that it has merit but does not fully meet PLOS Computational Biology's publication criteria as it currently stands. Therefore, we invite you to submit a revised version of the manuscript that addresses the points raised during the review process.

Please submit your revised manuscript within 60 days Feb 17 2025 11:59PM. If you will need more time than this to complete your revisions, please reply to this message or contact the journal office at ploscompbiol@plos.org. Please include the following items when submitting your revised manuscript:

We look forward to receiving your revised manuscript.

Kind regards,

Jonathan David Touboul

Academic Editor

PLOS Computational Biology

Marieke van Vugt

Section Editor

PLOS Computational Biology

**Additional Editor Comments (if provided):**

**Journal Requirements:**

**Reviewers' comments:**

Reviewer's Responses to Questions

**Comments to the Authors:**

Reviewer #1: This paper employs an impressive range of research modalities to address a central question in neuroscience: how cellular and receptor level phenomena can lead to brain-level behaviors. Significantly, it uses methods that span levels of analysis—tissue level computational modeling; gene profiling of receptor expression across brain regions; and conventional (gamma band) as well as more innovative (aperiodic power spectrum) analyses applied to MEG data—to arrive at its conclusions. The power of this approach is that it can produce non-obvious insights into how these micro-level effects give rise to emergent behaviors. While the N used in this study is relatively small, this is more than made up for by the fact that it is highly innovative methodologically—I feel that this is perhaps the most important contribution of this paper.

Two areas for improvement are as follows:

I. The use of the aperiodic power spectrum as an outcome measure is a very nice aspect of this paper, but I feel that it is underdeveloped. As the paper stands, I think that many readers may not appreciate its potential significance. As I am sure the authors are aware, it is referred to in various ways in the literature (“1/f noise”, “power law exponent”), and it is a phenomenon that occurs in many large complex systems (for example, [1]). The authors may wish to highlight this—or at a minimum, the literature that has examined this in human brain, and has speculated on its functional significance (for example [2]). Also, I feel that an explanation of this should not be buried in the Supplementary Materials, but included in the Methods and described in a way that provides some “intuition” for those less technically inclined.

II. The work presented is rigorous methodologically, but in its current form it comes across as something of a “technical exercise”. It could be improved by a greater discussion of its clear relevance to a clinical problem. Schizophrenia patients (SZ), of course, show deficits in gamma band activity. While in SZ, these tend to be decreases in driven (e.g., steady state auditory response) activity, and the current paper looks at gamma in the resting state, given the substantial literature, not delving into this seems like an oversight. Also, the glutamatergic hypothesis of schizophrenia (3) posits that glutamatergic transmission deficiencies are the underlying pathology in SZ, or at least one subtype of the illness. I realize that the authors have mentioned the potential relevance to SZ in passing. But I feel that greater development of this (e.g., how this work may shed light on an underlying cause of SZ, or illuminate how neuropathology may lead to functional deficits, or point the way to better treatments) would improve the paper and make it interesting to a wider audience.

(1) P. Bak, How nature works: the science of self-organized criticality (Copernicus, New York, NY, USA, 1996).

(2) B. J. He, Scale-free brain activity: past, present and future. Trends Cogn Sci 18, 480–487 (2014).

(3) J. T. Coyle, W. B. Ruzicka, D. T. Balu, Fifty Years of Research on Schizophrenia: The Ascendance of the Glutamatergic Synapse. Am J Psychiatry 177, 1119–1128 (2020).

Reviewer #2: Summary:

This work is a nice combination of approaches showing MEG changes in participants in response to ketamine, linking these effects to gene expression, and using computational models to uncover associated underlying microcircuit mechanisms. In general, while I find the work very interesting, I think that the gene expression analyses could benefit from being expanded and the simulations can be refined accordingly to better link with the gene expression findings and strengthen the validity of the simulations. Below I outline this and several other key points for consideration.

Major:

MEG-related

-If no change is seen in the slope of the aperiodic component in the lower frequency range (Fig S2), it seems like changes in gamma power may be driving the change in the aperiodic slope. So, is the change in gamma power and change in aperiodic slope just two measurements of the same effect? Can the authors demonstrate that the slope changes are not driven by gamma power entirely (e.g. using subtraction methods) to warrant including the aperiodic slope feature? It may also be helpful to see more examples of aperiodic fits in Fig S1 to see a range of aperiodic slopes.

-From the plots showing the power spectra in the aperiodic range, there does appear as though there may be ketamine-associated decreases in lower frequency peaks. Can the authors do a more complete power spectral decomposition of the periodic component (for both the MEG data and the simulated MEG) to quantify if there are changes in theta, alpha, beta, and gamma peaks after the aperiodic component is subtracted?

-What are the changes to oscillatory event waveforms in the gamma range following ketamine application (in the MEG recordings and in the model simulated MEG)? These are additional features that the authors could check whether the models with NMDA block can capture.

Gene expression-related

-Given the significant partial correlation of GRIN2D with aperiodic slope – in what cell types does GRIN2D tend to show higher expression? Also, have any previous experiments demonstrated that GluN2D has enhanced sensitivity to ketamine compared to other NMDA subunits?

-Line 293-294: “Furthermore, Ketamine is not only an NMDA-R antagonist but also targets dopaminergic, serotonergic and cholinergic receptors [48].” Can the authors answer this question by expanding their gene expression correlation analysis to include these receptor subunits? Similarly, inclusion of other cell type markers (e.g. L1 interneurons or pyramidal cell markers) would equally be informative and interesting.

-I am not clear on the details of the gene expression vs MEG data correlation analysis (fig 3 and lines 370-381 in the methods). My initial understanding was that each datapoint in the correlation analysis was a brain area (e.g. a centroid in Fig 3B) of mean MEG difference (KET-PLA) vs gene expression. On line 180 describing Fig 3B however it says “Brighter dots indicate higher correlation values”. How can single centroids possess single correlation values? I thought that the brightness represented difference estimates and gene expression values. Please clarify this and add color scales to fig 3B.

Simulation-related

-While the simulations showed that both PV and SST could be involved, the gene expression analysis only supported the involvement of PV. So how much SST is actually involved depends on how much NMDA modulates SST cells to begin with. It’s important to keep in mind that the models (at baseline) assume synapses with equal conductances from NMDA and AMPA so they could be over-estimating the effects of NMDA loss in any particular cell type. As far as I know NMDA activity contributions in human interneurons have not been measured and these models are not fit to any such data. With that in mind, I think this work could benefit from better using the gene expression-MEG correlation results to guide the key simulations to show in the main figures.

-Given the result of PV expression being correlated with larger ketamine-driven effects, it would relevant to check if increasing the PV cell count in the model increases the effects of NMDA block on simulated MEG.

-I think more conditions in the model could be explored to find realistic contexts where the aperiodic slope is flattened. From the tests that are already shown, decreased NMDA to PV is a good candidate for causing an elevated PSD at higher frequencies (and is supported by the gene expression analysis), and decreased NMDA to Pyr is a good candidate for counteracting the elevated PSD at lower frequencies. Why not explore combinations of just those two targets to see if together they flatten the aperiodic slope?

-In Mazza et al 2023 reduced inhibition from SST, but not PV, led to a reduced aperiodic slope. Although the mechanism here is slightly different (NMDA input reduction vs inhibitory output reduction), it’s worth discussing why these differences (whether it’s mechanistic-related, or method-related) may be leading to different outcomes.

-A 2 s simulation duration seems short for computing the power spectra, which could affect lower frequency amplitudes of the aperiodic component (and thus the slope). Mazza et al 2023 used simulations of 28 s for the power spectral analysis. Given the focus of this paper on higher frequency oscillations and aperiodic activity, it is unclear if analyzing short simulations will impact the results, but the aperiodic slope is also modulated by lower frequency ranges. Can the authors demonstrate that their 2s simulations generate sufficient power spectra compared to example longer simulations before and after NMDA block?

-It’s worth considering some conceptual differences with the model considerations in Yao et al 2023 where depression was simulated via reduced SST inhibitory output conductance. Ketamine is a rapid antidepressant and these results suggest that reducing NMDA drive to SST interneurons may underlie ketamine effects. Thus a loss of SST interneuron contributions is expected in both depression and when applying a rapid antidepressant. Can the authors discuss this counterintuitive result?

-While the experiments specifically look at sub-anesthetic administration of ketamine, it is unclear if the simulated NMDA blocks correspond to such a dosage – effects appeared most apparent with PV and SST starting at a 40% reduction and up to 60% and 100%, respectively. Can the authors provide a rationale for why these values represent the action of ketamine during sub-anesthetic dose administration?

-The effects of NMDA activity reductions on power spectral metrics in the model are not quantitatively shown. Can the authors show plots linking the mechanistic changes in the model with changes in the power spectral metrics (gamma power and aperiodic slope – like fig 2C & F)? As well, can the authors demonstrate what percent changes in the model lead to similar percent changes in power spectral features seen experimentally during ketamine application?

-Fig 6 appears to be taken directly from Yao et al 2022 and Mazza et al 2023. While this is acknowledged in the caption, panel C is a bit misleading given the differences in simulation time, EEG vs MEG methods, and analysis methods. I would suggest replacing panel C with something that illustrates the MEG simulation method that they added (lines 408-414). Adding a panel to better illustrate the key model manipulations would also be helpful. I also find that having this type of figure before presenting the results (or combined with the results figures) generally makes it easier for the reader (otherwise one must skip to the last figure to understand what’s going on).

Minor:

-The organization of figs 1 and 2 with the corresponding results text is non-sequential – can the authors re-organize (either the text or the plots) for it to read better (i.e. section 1 -> fig 1, section 2 -> fig 2)?

-Fig 2D looks at the aperiodic component, but this looks like it’s just the averaged raw PSDs being plotted in that frequency range. Can the authors also plot the mean aperiodic fits?

-In Fig S2 it would also be beneficial to show the mean aperiodic fits.

-Also in Fig S2, for better visibility, can the authors zoom in on the ranges in the 7-30 Hz and 30-80 Hz cases rather than highlighting them on the larger scale (or show the zoomed in versions as insets)?

-Fig S2: It’s interesting that significant ketamine effects on aperiodic slope were only seen in subcortical regions when the whole frequency range was considered – can the authors discuss this result and give more context to it?

-It’s not clear why VIP and Pyr neurons were only analyzed for a -80% and -80%/-40% decreases in NMDA whereas PV and SST had multiple levels explored.

-In the methods, when detailing NMDA reductions in specific cell types, are the authors referring to NMDA conductance when they say “NMDA activity”? Please clarify this detail.

-Lines 388-389: The data provenance descriptions and refs for the Yao et al model are not quite correct. All cell types in this model were fit to human in vitro spiking data. Key connections between cell types (Pyr to Pyr, Pyr to SST, SST to Pyr, Pyr to PV and PV to Pyr) were also fit to human data. Data from ref 67 (Obermayer et al 2018) was used to fit Pyr to SST and SST to Pyr connections, not Pyr model spiking. Data from refs 68-70 (Ma et al 2006, Prönneke et al 2015, and Zurita et al, 2018) were not used to fit the models at all, but as sources for cross-validating the models after the optimizations to check if the model features were within population variance estimates (since the interneurons were fit to human single-cell feature values and not mean population feature values). Please consult table S1 in Yao et al 2022, for a full list of data provenance, and correct these details where applicable.

-Fig S3 caption appears to have a typo: “PV -40%, SST -20%, PV -15%, pyramidal neurons -5%” I think should say “PV -40%, SST -20%, VIP -15%, pyramidal neurons -5%”.

Reviewer #3: The study investigates how ketamine influences the balance of excitation and inhibition in cortical microcircuits, particularly focusing on the roles of parvalbumin and somatostatin interneurons. By analyzing MEG data from 12 participants, the authors explore the changes in gamma-band power and the correlation with gene expression.

Key Findings:

1. Increased Gamma-Band Power: The authors report a significant increase in gamma-band power following Ketamine administration, particularly in prefrontal and central regions.

2. Flatter Aperiodic Slope: The observation of a flatter aperiodic slope is noteworthy, as it suggests increased excitation.

3. Interneuron Dynamics: The computational modeling aspect of the study is a significant contribution, as it attempts to link NMDA receptor dysfunctions in specific interneurons to observed changes in gamma-band power. However, the model's inability to account for changes in the aperiodic slope raises questions about its completeness.

Major

The authors focus their attention on the gamma band oscillations (30-90 Hz). As decreases in lower beta-band oscillations have also been shown to correlate with psychotomimetic effects of ketamine (DOI: 10.1007/s00213-022-06272-9), it is suggested that the authors also include analysis of e.g. delta/theta/alpha/beta and potentially HFO bands to support the aperiodic and gamma-band findings. If gamma-band is sufficient for the main red thread of the paper, then this information could be presented as supplementary information. However, in figure 2D, a clear reduction in 10-20 Hz power is visible suggesting that this frequency range may also be involved in the reduction in the aperiodic slope. How was the balance point set at 30 Hz and would the conclusion have been different had it been set at e.g. 20 or 25 Hz?

The “disinhibition hypothesis” or E/I balance is controversial and has been challenged from many sides. The Homayoun et al 2007 paper is a popular paper to cite when explaining ketamine effects, however in that paper they investigated MK-801 and not ketamine. If the focus is on subunit specific effects it would be good to take into account that MK-801 has different NMDAR subunit selectivity than ketamine (which is really not very selective, DOI: 10.1523/JNEUROSCI.3703-08.2009). One excellent review discusses pros and cons for the disinhibition or direct inhibition theories. It may be advantageous to take this in, as the computational model does not account for the changes in slope seen in the MEG-data. Furthermore, in rodent experiments in awake and freely moving animals both ketamine and PCP induced increased firing from pyramidal cells without reduction in the activity of local inhibitory interneurons. This further challenges the traditional "disinhibition hypothesis" that suggests NMDA receptor antagonists primarily work by inhibiting GABAergic interneurons to enhance pyramidal neuron activity. In the same study, increases in gamma and HFO were found suggesting that an increase in gamma-band power is not dependent on local disinhibition (https://doi.org/10.1016/j.neuropharm.2019.107745).

Correlation with gene expression profiles. It would strengthen the paper if the authors also showed correlations to other frequency bands that are affected by ketamine – or that there are none because the gamma band is the more relevant one.

I have no reason to think that the quality of data or the analysis should not be of high standard and trust that the study has been expertly performed in any way. I believe that the study will be of great interest to the field, especially if the introduction and discussion are adjusted and the additional data from adjacent frequency bands will be included. d

**Have the authors made all data and (if applicable) computational code underlying the findings in their manuscript fully available?**

Reviewer #1: Yes

Reviewer #2: Yes

Reviewer #3: Yes

PLOS authors have the option to publish the peer review history of their article (what does this mean? ). If published, this will include your full peer review and any attached files.

**Do you want your identity to be public for this peer review?** For information about this choice, including consent withdrawal, please see our Privacy Policy .

Reviewer #1: No

Reviewer #2: No

Reviewer #3: No

**Figure resubmission:**
---

## [Decision Letter · Decision Letter 1]

PCOMPBIOL-D-24-01769R1

Computational Modeling of Ketamine-Induced Changes in Gamma-Band Oscillations: The Contribution of Parvalbumin and Somatostatin Interneurons

PLOS Computational Biology

Dear Dr. uhlhaas,

Thank you for submitting your manuscript to PLOS Computational Biology. After careful consideration, we feel that it has merit but does not fully meet PLOS Computational Biology's publication criteria as it currently stands. Therefore, we invite you to submit a revised version of the manuscript that addresses the points raised during the review process.

Please submit your revised manuscript within 30 days Jun 16 2025 11:59PM. If you will need more time than this to complete your revisions, please reply to this message or contact the journal office at ploscompbiol@plos.org. Please include the following items when submitting your revised manuscript:

We look forward to receiving your revised manuscript.

Kind regards,

Jonathan David Touboul

Academic Editor

PLOS Computational Biology

Marieke van Vugt

Section Editor

PLOS Computational Biology

**Additional Editor Comments :**

As you will see, two of the Reviewers find your responses compelling and recommended acceptance of the paper. Reviewer 2 also requested to consider going deeper on their initial points 1 and 7. I wanted to give you an opportunity to consider these comments and have an opportunity to think about these before we can accept your paper. In particular, I find the first point of the reviewer interesting to explore and comment, even possibly, as they suggest, as a possible caveat.

**Journal Requirements:**

1) We note that your S1 File. AAL region abbreviations is uploaded twice in the online submission form with two different labels as S1_File.docx and S2_File.docx . Please remove any unnecessary files from your revision, and make sure that only those relevant to the current version of the manuscript are included.

2) Thank you for stating “The MEG and clinical data is not made publicly available as not explicit permission was obtained to share the data with third parties.” Please inform us if you could provide deidentified data. If so, please upload the minimal anonymized data set necessary to replicate your study findings to a stable, public repository and provide us with the relevant URLs, DOIs, or accession numbers. For a list of recommended repositories, please see https://journals.plos.org/plosone/s/recommended-repositories. You also have the option of uploading the data as Supporting Information files, but we would recommend depositing data directly to a data repository if possible.

3) Please amend your detailed Financial Disclosure statement. This is published with the article. It must therefore be completed in full sentences and contain the exact wording you wish to be published.

4) Please include the affiliation of the author (Peter J. Uhlhaas) in the online submission form.

**Reviewers' comments:**

Reviewer's Responses to Questions

Reviewer #1: The authors have nicely responded the criticisms that I have raised. This is a highly methodologically novel paper that is appropriate for publication.

Reviewer #2: Summary:

I am satisfied with most of the authors responses. However, I have two remaining points that I would like the authors to consider addressing further.

Major:

Response to point 1:

In their response the authors try to make the case that the two measures are not measures of the same effect. However, the points that they give only seem to argue for them being driven by a similar effect. For example, they mentioned that there is a strong correlation between aperiodic slope and gamma power. They say that, albeit not a perfect relationship, it is not possible to disentangle the effect and show that gamma power is not driven by aperiodic slope. This only convinces me further that the two metrics are likely two measurements of the same effect. I’d suggest that this should be addressed more clearly in their manuscript, even if just as a caveat.

The correlation might actually be due to a peak on the upper end of the frequency range – does stopping at 80/90 Hz cut into gamma peaks, such that gamma peaks end up affecting the flatness of the aperiodic component? In fact, the upward inflection in the right tail of the separated periodic component (from the point 2 checks) suggests that there is a peak being cut near 80-90 Hz. Does extending the range past this peak (or even decreasing to before the peak) change the findings on aperiodic slope? And if not, is the correlation between aperiodic slope and gamma amplitude still present?

Responses to point 2-6: I am satisfied with the additional checks and clarifications on these points.

Response to point 7:

Regarding my point on using the gene expression-MEG analysis to guide simulations - if it was expected that SST and VIP expression contribute to ketamine-driven changes in gamma and aperiodic slope, please discuss possible reasons why they were not correlated with these changes. That is, discuss whether there are methodological issues that might occlude these correlations or if these are likely true negative results. Although the authors provide some justifications (in the intro and discussion) for simulating NMDA changes to SST and VIP interneurons, because these were not supported by the expression-MEG correlation results, the simulations come across as being disjointed from the gene expression-MEG correlation results.

Response to points 8-15: I am satisfied with these checks, changes, and explanations.

Minor:

Response to point 16: I think exactly the opposite – to me, this structure only impedes the reader’s understanding of the key findings. However, this is a stylistic preference and very minor so I will not push further.

Response to points 17-22: I am satisfied with how these points were addressed.

Reviewer #3: Congratulations with a skillfully performed well written manuscript. I have no further comments.

**Have the authors made all data and (if applicable) computational code underlying the findings in their manuscript fully available?**

Reviewer #1: Yes

Reviewer #2: Yes

Reviewer #3: None

PLOS authors have the option to publish the peer review history of their article (what does this mean? ). If published, this will include your full peer review and any attached files.

**Do you want your identity to be public for this peer review?** For information about this choice, including consent withdrawal, please see our Privacy Policy .

Reviewer #1: No

Reviewer #2: No

Reviewer #3: No

**Figure resubmission:**
---

## [Editor Report · Decision Letter 2]

Dear Professor uhlhaas,

We are pleased to inform you that your manuscript 'Computational Modeling of Ketamine-Induced Changes in Gamma-Band Oscillations: The Contribution of Parvalbumin and Somatostatin Interneurons' has been provisionally accepted for publication in PLOS Computational Biology.

Best regards,

Jonathan David Touboul

Academic Editor

PLOS Computational Biology

Marieke van Vugt

Section Editor

PLOS Computational Biology

---

## [Editor Report · Acceptance letter]

PCOMPBIOL-D-24-01769R2

Computational Modeling of Ketamine-Induced Changes in Gamma-Band Oscillations: The Contribution of Parvalbumin and Somatostatin Interneurons

Dear Dr Uhlhaas,

I am pleased to inform you that your manuscript has been formally accepted for publication in PLOS Computational Biology. Your manuscript is now with our production department and you will be notified of the publication date in due course.

With kind regards,

Anita Estes
